Transcriptomic analysis reveals candidate genes for male sterility in Prunus sibirica

Chen Jianhua
Xu Hao
Zhang Jian
Dong Shengjun dsj928@163.com
Liu Quangang liuquangang007@126.com
Wang Ruoxi
College of Forestry, Shenyang Agricultural University , Shenyang , Liaoning , China
Eamens Andrew
Electronic publication date: 2021 Oct 20
Publication date: 2021
Volume: 9
Electronic Location ID: e12349
Received 2021 Mar 23; Accepted 2021 Sep 29
Copyright: ©2021 Chen et al.
Copyright year: 2021
Copyright holder: Chen et al.
License: This is an open access article distributed under the terms of the Creative Commons Attribution License, which permits unrestricted use, distribution, reproduction and adaptation in any medium and for any purpose provided that it is properly attributed. For attribution, the original author(s), title, publication source (PeerJ) and either DOI or URL of the article must be cited.
License URL: https://creativecommons.org/licenses/by/4.0/

Keywords: Prunus sibirica, Male sterility, Microstructure, Transcriptome sequencing, RT-qPCR

Funding: National Key Research and Development Program of China SQ2019YFD100071 This work was supported by the National Key Research and Development Program of China (SQ2019YFD100071). The funders had no role in study design, data collection and analysis, decision to publish, or preparation of the manuscript.

==============================
Background

The phenomenon of male sterility widely occurs in Prunus sibirica and has a serious negative impact on yield. We identified the key stage and cause of male sterility and found differentially expressed genes related to male sterility in Prunus sibirica, and we analyzed the expression pattern of these genes. This work aimed to provide valuable reference and theoretical basis for the study of reproductive development and the mechanisms of male sterility in Prunus sibirica.

Method

The microstructures of male sterile flower buds and male fertile flower buds were observed by paraffin section. Transcriptome sequencing was used to screen genes related to male sterility in Prunus sibirica. Quantitative real-time PCR analysis was performed to verify the transcriptome data.

Results

Anther development was divided into the sporogenous cell stage, tetrad stage, microspore stage, and pollen maturity stage. Compared with male fertile flower buds, in the microspore stage, the pollen sac wall tissue in the male sterile flower buds showed no signs of degeneration. In the pollen maturity stage, the tapetum and middle layer were not fully degraded, and anther development stopped. Therefore, the microspore stage was the key stage for anther abortion , and the pollen maturity stage was the post stage for anther abortion. A total of 4,108 differentially expressed genes were identified by transcriptome analysis. Among them, 1,899 were up-regulated, and 2,209 were down-regulated in the transcriptome of male sterile flower buds. We found that “protein kinase activity”, “apoptosis process”, “calcium binding”, “cell death”, “cytochrome c oxidase activity”, “aspartate peptidase activity”, “cysteine peptidase activity” and other biological pathways such as “starch and sucrose metabolism” and “proteasome” were closely related to male sterility in Prunus sibirica. A total of 331 key genes were preliminarily screened.

Conclusion

The occurrence of male sterility in Prunus sibirica involved many biological processes and metabolic pathways. According to the results of microstructure observations, related physiological indexes determination and transcriptome analysis, we reveal that the occurrence of male sterility in Prunus sibirica may be caused by abnormal metabolic processes such as the release of cytochrome c in the male sterile flower buds, the imbalance of the antioxidant system being destroyed, and the inability of macromolecular substances such as starch to be converted into soluble small molecules at the correct stage of reproductive development, resulting in energy loss. As a result, the tapetum cannot be fully degraded, thereby blocking anther development, which eventually led to the formation of male sterility.

Introduction

Prunus sibirica is a shrub or small tree that belongs to the family Rosaceae. It is mostly found in the wild and semi-wild state, and most of Prunus sibirica are self-incompatible. There is a common phenomenon of interspecific hybridization in Prunus sibirica, so the germplasm resources are abundant. Prunus sibirica is widely cultivated in China, and its economic benefits are generally recognized by those who reside in the areas where this species is commercially cultivated (Wan et al., 2015). However, the low and unstable yield, which is caused by many factors, such as frost damage during flowering stage, low fruit setting rate, self-incompatibility, male sterility and pistil abortion, has become the main bottleneck which is hindering the continued development of the Prunus sibirica industry (Wang et al., 2014). Among them, the differentiation of male organs directly affects the yield of Prunus sibirica. At present, the key stage and cause of male sterility in Prunus sibirica have not been determined.

Plant male sterility is widespread in nature. At present, research conducted on different plant species, including apricot (Badenes et al., 2000), Prunus mume (Yaegaki et al., 2003), Prunus salicina (Radice et al., 2008), poplar (Liu et al., 2019a; Liu et al., 2019b), Camellia crassicolumna (Jiang et al., 2020), Brassica napus (Du et al., 2016), and tobacco (Liu et al., 2020a; Liu et al., 2020b), from the aspects of cytology, physiology and biochemistry, and molecular biology have been conducted in order to explore the mechanism of plant male sterility. Plant male sterility can usually be classified into several types such as abnormal meiosis of pollen mother cells (Nonomura et al., 2003; Zhou et al., 2011), abnormal metabolism of callose (Wan et al., 2011), early or late degeneration of tapetum cells (Jung et al., 2005; Li et al., 2006), abnormal development of pollen wall (Shi et al., 2011), and the failure of anther dehiscence (Steiner-Lange et al., 2003). The tapetum plays an important role in pollen development, which provides nutrients for the developing pollen (Pacini, 2010; Gómez, Talle & Wilson, 2015). The middle layer cells are located between the tapetum and endothecium, and play an important role in the differentiation and function of tapetum cells (Ma et al., 2007; Roque et al., 2007). In recent years, with the rapid development of high-throughput sequencing technology, specifically transcriptomic sequencing, has been widely used in the study of male sterility in herbs or cereals such as Salvia miltiorrhiza Bunge (Yu et al., 2021), wheat (Liu et al., 2020a; Liu et al., 2020b), Cucumis melo (Dai et al., 2019), onion (Yuan et al., 2018) and woody plant such as Vernicia fordii (Liu et al., 2019a; Liu et al., 2019b), and Citrus suavissima (Zhang et al., 2018a; Zhang et al., 2018b). However, to our knowledge, no study has used a transcriptomics approach to investigate male sterility in Prunus.

By using paraffin section, transcriptome sequencing and quantitative real-time PCR (RT-qPCR), this study clarified the critical stage and cause of male sterility in Prunus sibirica. The differentially expressed genes in male sterile flower buds and male fertile flower buds were analyzed, and the mechanism of male sterility was explored at the level of microstructure and gene transcription level. This work aims to provide a scientific basis for the study of the mechanism of male sterility in Prunus sibirica, and also lay the foundation for selecting sterile materials and creating high-yield and stable product varieties through use of genetic engineering technology.

Materials & Methods

Plant materials

The eight-year-old Prunus sibirica clones were selected as the experimental material, which were cultivated in the Prunus sibirica germplasm resource nursery of Shenyang Agricultural University (Beipiao, Liaoning, China). Based on our research team’s multi-year investigation of the inflorescence, florescence, male sterile flower buds of clone No. 1 were selected as the experimental group, and male fertile flower buds of clone No. 60 were selected as the control group. The anthers of clone No. 1 were aborted thoroughly, there was no phenomenon of pollen dispersal, and the sterility was stable. The stamens of clone No. 60 developed completely, and can release abundant mature pollen after flowering, and this clone can pollinate and bear fruits as well. Flower bud samples of male sterile clone and male fertile clone were collected from the end of July in 2018 to the beginning of April in 2019. Samples were taken every 10 to 15 days. Each time, approximately 10 to 20 complete flower buds free of any observable disease symptoms or insect pests were selected from the upper part of the fruiting branches outside the canopy on the sunny side of the plant. The samples were collected and quickly stored in FAA fixative to create paraffin sections of flower buds. At the same time, three biological replicates of flower buds were collected and rapidly transferred into liquid nitrogen (−196 °C). The samples were taken back to the laboratory and stored in an ultra-low temperature freezer (−80 °C) for RNA extraction. MSFB and MFFB were used to represent male sterile clone No. 1 and male fertile clone No. 60, respectively. MSFB_1, MSFB_2 and MSFB_3 were used to represent the experimental group, and MFFB_1, MFFB_2 and MFFB_3 were used to represent the control group.

Microstructure observation

Paraffin section technology was used to make sections (Xu et al., 2008). The microstructure of male sterile flower buds and male fertile flower buds from each sampling point were observed and photographed with an inverted fluorescent microscope (Zeiss Axio Vert.A1).

Determination of physiological indicators

The anthrone colorimetric method was used to determine the contents of soluble sugar and starch (Gao, 2005). Coomassie Brilliant Blue G-250 staining was used to determine the contents of soluble protein (Qu et al., 2006).

RNA extraction and detection

Total RNA was extracted from male sterile flower buds and fertile flower buds by using an RNA extraction kit (Tiangen Biotech Co., Ltd., Beijing, China) according to the manufacturer’s instructions. Post total RNA extraction, a 1.0% (w/v) agarose gel electrophoresis was used to detect whether the extracted RNA was degraded or of appropriate quality. A NanoPhotometer spectrophotometer (IMPLEN, CA, USA) was also employed to further assess the purity of each RNA sample, and an Agilent Bioanalyzer 2100 system (Agilent Technologies, CA, USA) was used to detect the concentration and integrity of each RNA sample.

Construction of sequencing library and Illumina sequencing

The sequencing library was constructed by using the NEBNext® Ultra™ RNA Library Prep Kit for Illumina® (NEB, USA) sequencing. First, poly (A) mRNA was purified from total RNA with Oligo(dT) magnetic beads. Fragmentation was carried out using divalent cations in NEBNext First Strand Synthesis Reaction Buffer (5X). Fragmented mRNA was taken as the template. The first strand cDNA was synthesized using a random hexamer primer and M-MuLV Reverse Transcriptase (RNase H-). The synthesis of second strand cDNA was subsequently performed using DNA polymerase I and RNase H. The obtained double-stranded cDNAs were end-repaired. A tail was added, and a sequencing connector was attached. In order to select cDNA fragments with 250∼300 base pairs (bp) of length, the library fragments were purified with AMPure XP beads (Beckman Coulter, Beverly, USA). The purified cDNA was amplified by PCR, the PCR products were purified again by AMPure XP beads, and finally the cDNA libraries were obtained. The constructed libraries were quantified by using a Qubit 2.0 Fluorometer, and then the library quality was detected by using an Agilent 2100 bioanalyzer. To ensure the quality of the libraries, RT-qPCR was used to accurately quantify the effective concentration of the libraries. Meanwhile, the clustering of the samples was performed on a cBot Cluster Generation System using TruSeq PE Cluster Kit v3-cBot-HS. After cluster generation, the library preparations were sequenced on an Illumina HiSeq platform and 150 bp paired-end reads were generated.

Transcriptome sequencing data analysis

The raw image data obtained from sequencing were converted to sequence reads using CASAVA. Then, the raw data were filtered by R language, and finally the clean reads for subsequent analysis were obtained. Trinity v2.4.0 program was used to stitch and assemble clean reads of all samples (Grabherr et al., 2011). Redundancy was removed by clustering with Corset hierarchy. The longest transcript of each gene was selected as the ‘unigene’ sample for subsequent analysis. The transcript sequence obtained by splicing with Trinity was used as a reference sequence. RSEM software (version v1.2.15) (bowtie2, mismatch = 0) was used to compare clean reads of each repeated sample with reference sequences (Li & Dewey, 2011), and the readcount of each gene was counted and compared. Fragments per kilobase of exon per million mapped fragments (FPKM) was used to standardize the readcount of genes.

Gene functional annotation

Gene function was annotated based on the following databases: Nr: Nr library includes the protein coding sequence of GenBank gene and protein sequence of PDB, SwissProt, PIR and PRF database (diamond v0.8.22, e-value = 1e−5). Nt: Nt library includes the nucleic acid sequence of GenBank, EMBL and DDBJ (NCBI blast 2.2.28+, e-value = 1e−5). Pfam: Pfam based on the conservation of protein domains, Pfam library annotates the protein family of genes through the establishment of an HMM statistical model (HMMER 3.0 package, e-value = 0.01). KOG: based on gene homology relationship, KOG library classifies gene function according to evolution relationship (diamond v0.8.22, e-value = 1e−3). SwissProt: SwissProt belongs to an annotated protein sequence database, including protein function and transcription Post-modification, special site and region information (diamond v0.8.22, e-value = 1e−5). KEGG: KEGG library belongs to a database for systematic analysis of gene product functions and metabolic pathways (KASS, e-value = 1e−10). GO: GO library belongs to a set of international standardized classification system for the description of gene function, which can be divided into three categories: biological process, molecular function and cellular component (Blast2GO v2.5, e-value = 1e−6).

Differentially expressed gene analysis

The DEGseq R package (1.12.0) was used for differential expression analyses. The P-value was calculated on the basis of a negative binomial distribution model. P-values were adjusted using the Benjamini–Hochberg method. Genes with an adjusted P-value < 0.05 and log2(Fold change) > 1 were considered as differentially expressed.

Gene ontology (GO) and KEGG enrichment analysis

Based on the hypergeometric distribution, GOseq R package (1.10.0) was used for GO enrichment analysis, and KOBAS (v2.0.12) was used for KEGG pathway enrichment analysis.

Quantitative real time PCR (RT-qPCR) analysis

RNA was extracted as described above. The first-strand cDNA was synthesized using EasyScript One-Step gDNA Removal and cDNA Synthesis SuperMix (TransGen Biotech, China). The 18SrRNA gene was used as an internal reference gene (Table S1). RT-qPCR was performed with SYBR Green I method. The reaction system was 20 µL, including 10 µL of 2  × SuperReal PreMix Plus, 0.6 µL of 0.3 µmol/L forward primer, 0.6 µL of 0.3 µmol/L reverse primer, 1.0 µL of cDNA template, 2.0 µL of 50  × ROX Reference Dye, and 5.8 µL ddH2O. The RT-qPCR was performed on the Applied Biosystems Step One Plus system, and the experiments were carried out in three replications. The PCR program was as follows: initial denaturation at 95 °C for 15 min, denaturation at 95 °C for 10 s, annealing at 60 °C and extension for 32 s, for 40 cycles. The 2−ΔΔCt method was used to calculate the relative expression. The correlation coefficient between transcriptome sequencing and RT-qPCR was analyzed using SPSS 22.0 software.

Figure 1 The microstructure of male sterile flower buds (A–D) and male fertile flower buds (E–H) at different developmental stages in Prunus sibirica.

(A, E) Sporogenous cell stage; (B, F) tetrad stage; (C, G) microspore stage; (D, H) anther maturity stage. Sp: sporogenous cells; Ps: pollen sac; T: tapetum; Tds: tetrads; Ep: epidermis; En: endothecium; ML: middle layer; MSp: microspores; A, D: sporogenous cell stage; B, F: tapetum stage.

Results

Microstructural characteristics of male sterile flower buds and male fertile flower buds of Prunus sibirica

To identify the key period and characteristics of male sterility in Prunus sibirica, the microstructures of male sterile flower buds and male fertile flower buds at different developmental periods were observed by paraffin section (Fig. 1). The results showed that anther development can be divided into four stages, namely sporogenous cell stage, tetrad stage, microspore stage, and pollen maturity stage. At the sporogenous cell stage (Figs. 1A, 1E) and tetrad stage (Figs. 1B, 1F), no significant difference was observed in anther development between male sterile flower buds and male fertile flower buds of Prunus sibirica. The sporogenous cells located in the four corners of the anther differentiate into multiple microspore mother cells. These cells produce microspores through meiosis, and they are surrounded by callose to form tetrads. When flower buds reach the microspore stage, microspores are released from tetrads with the degradation of the callose. In male fertile flower buds, the tapetum and middle layer cells in the pollen sac wall were degraded, and which rendered the border between the two difficult to observe. At the pollen maturity stage, the tapetum and middle layer cells disappeared completely. Only the endothecium and epidermis were left in the pollen sac wall (Figs. 1G, 1H). In male sterile flower buds however, the pollen sac wall tissue remained completely at the microspore stage, with any signs of degradation failing to be observed. At the pollen maturity stage, the tapetum and middle layer cells were not degraded sufficiently, and the development of pollen was blocked, which eventually led to male sterility (Figs. 1C, 1D). Therefore, the abnormal anther development was the cause of male sterility in Prunus sibirica. The microspore stage was the key stage of anther abortion, and the pollen maturity stage was the post stage of anther abortion.

Total RNA extraction and detection

The total RNA concentrations of the six samples ranged from 452 ng µL−1 to 670 ng µL−1. The RNA integrity number values were all close to 10 (Table S2). These results indicated that the extracted RNA had good integrity, high purity, and no obvious degradation, which met the quality requirements of sequencing library construction.

Six cDNA libraries of male sterile flower buds and male fertile flower buds at the microspore stage of abortion were sequenced on an Illumina sequencing platform using double-ended sequencing. After filtering and quality control, a total of 40,030,500∼61,826,716 clean reads were obtained. The clean reads rate was between 98.02% and 98.45%. The total number of clean bases was 45.80 Gb, the sequencing quality represented by Q30 percentage was over 94%, and the GC content was between 45.65% and 45.88%. The transcripts assembled by Trinity were used as the reference transcriptome, and the clean reads of each sample were mapped to the reference sequences. The alignment proportion of each sample was greater than 80% (Table S3). The average length, median length and N50 of the assembled unigenes were 1,568 bp, 1,044 bp and 2,520 bp, respectively (Table S4). All the above results showed that the quality and accuracy of the sequencing data were sufficient for further analysis.

Trinity software was used to assemble the clean reads and obtain non-redundant unigenes, and a total of 34,377 unigenes were obtained. The gene function of the obtained unigenes were annotated in seven databases (Nr, Nt, KEGG, SwissProt, Pfam, GO, and KOG), and it was found that a total of 27,798 unigenes were annotated, accounting for 80.86% of the number of originally identified unigenes. The number of unigenes that were annotated successfully in all seven databases was 3,283, accounting for 9.54% (Table S5).

Analysis of the differentially expressed genes

The expressed genes with an adjusted P-value < 0.05 and log2(Fold change) > 1 were designated as differentially expressed genes. Differences in gene expression in male sterile flower buds and male fertile flower buds were compared and analyzed. A total of 4,108 genes were differentially expressed with 1,899 up-regulated genes and 2,209 down-regulated genes in male sterile flower buds (Fig. S1).

GO functional enrichment analysis of differentially expressed genes

GO functional classification analysis of the differentially expressed genes between male sterile flower buds and male fertile flower buds was conducted. The results are shown in Fig. 2. Among the biological processes, differential genes were mainly enriched in the processes of “cellular process”, “metabolic process”, and “single-organism process”. In cellular component category, the differential genes were mainly distributed in “cells”, “cell part”, “cell composition”. In the molecular function category, the differential genes were mainly enriched in “binding”, “catalytic activity”, and “heterocyclic compound binding”.

Figure 2 GO functional classification of differentially expressed genes.

Based on the results of GO enrichment analysis, eight GO functional subclasses with significant enrichment (adjusted P-value < 0.5) were selected (Fig. 3), including “DNA integration” in biological process and “ADP binding”, “heme binding”, “tetrapyrrole binding”, “iron ion binding”, “oxidoreductase activity”, “acting on paired donors with incorporation or reduction of molecular oxygen”, “oxidoreductase activity”, and “terpene synthase activity” in molecular function. The results indicated that these functional categories played a central role in the occurrence of male sterility in Prunus sibirica.

Figure 3 GO significant enrichment analysis of differentially expressed genes.

According to microstructure observations, the insufficient degradation of tapetal cells at the pollen maturity stage was one of the causes of male sterility in Prunus sibirica. The degradation process of tapetal cells belongs to programmed cell death. Therefore, the genes involved in the categories of “protein kinase activity”, “apoptosis process”, “calcium binding”, “cell death”, “cytochrome c oxidase activity”, “aspartate peptidase activity” and “cysteine peptidase activity” may be the key genes that regulate the male sterility of Prunus sibirica. Among them, a total of 296 related genes were enriched, and the detailed information of the 296 genes is listed in Table S6.

Hierarchical clustering was performed on key differentially expressed genes of male sterility in Prunus sibirica, which were screened on the basis of GO function enrichment analysis, and the heat map was drawn (Fig. S2). The column represents the expression of the same gene in different samples, and the horizontal row represents the expression of different genes in the same sample. The differences between individual samples at the transcriptome level can be preliminarily depicted by the use of a cluster map, and the differentially expressed genes can be divided into four categories (Category 1, Category 2, Category 3 and Category 4). The detailed information of differentially expressed genes in GO functional enrichment analysis is listed in Table S6.

KEGG pathway enrichment analysis of differentially expressed genes

To explore the main metabolic pathways in which the differentially expressed genes are involved, KEGG pathway cluster analysis was conducted on differentially expressed genes between male sterile flower buds and male fertile flower buds of Prunus sibirica. The differentially expressed genes were mapped to 215 biological pathways, and 20 of them were significantly enriched (Fig. 4). They mainly included “plant-pathogen interaction”, “monoterpenoid biosynthesis”, “protein processing in endoplasmic reticulum”, “flavonoid biosynthesis”, “Stilbenoid, diarylheptanoid and gingerol biosynthesis”, “phenylpropanoid biosynthesis”, “Vitamin B6 metabolism”, “glycolysis/gluconeogenesis”, and “diterpenoid biosynthesis”. These pathways might play a key role in directing the occurance of male sterility in Prunus sibirica.

Figure 4 KEGG pathway significant enrichment analysis of differentially expressed genes.

The metabolism of macromolecular nutrients plays a key role in the formation of male sterility in plants. In this study, the differences of soluble sugar content, starch content, and soluble protein content between male sterile flower buds and male fertile flower buds were analyzed. The results showed that during the critical stage of anther abortion, the soluble sugar content, starch content and soluble protein content in male sterile flower buds were significantly lower than those in male fertile flower buds at the same stage. During the post-abortion stage, the soluble sugar content and soluble protein content in male sterile flower buds were significantly lower than those in male fertile flower buds, whereas the starch content was significantly higher than that of male fertile flower buds (Fig. 5, Table S7).

Figure 5 Analysis of related physiological indexes between MSFBs and MFFBs in Prunus sibirica.

Combined with the results of KEGG pathway enrichment analysis, we found that “starch and sucrose metabolism” and “proteasome” pathways were enriched separately, and a total of 35 differentially expressed genes were assigned to these pathways (Fig. S3). This result strongly suggests that the differentially expressed genes played an important regulatory role in the occurrence of male sterility in Prunus sibirica. The detailed information of differentially expressed genes in KEGG pathway enrichment analysis is listed in Table S8.

RT-qPCR analysis

To verify the accuracy of the transcriptome sequencing data, 14 DEGs were selected and used for RT-qPCR verification. The detailed information of the 14 DEGs is listed in Table S9. The Ct value is listed in Table S10. The RT-qPCR results were largely consistent with the transcriptome data (Fig. 6), and the correlation coefficient was determined to be 0.964. The results showed that the accuracy of transcriptome sequencing was high, which can be used to analyze the dynamic changes to the expression of genes related to male sterility in the flowers of Prunus sibirica, thus revealing the molecular mechanism of male sterility.

Figure 6 RT-qPCR verification for transcriptome sequencing results.

Discussion

In the studies of male sterility of Prunus conducted to date, Lillecrapp, Wallwork & Sedgley (1999) found that the anthers of ‘Trevatt Blue’ apricot contained degradedmicrospores, with some failure in tapetal breakdown. Further, Badenes et al. (2000) found a marker (M4-650) linked to male fertility traits in apricot using RAPD markers combined with the BSA method. In addition, Yaegaki et al. (2003) found that genes related to the male sterility of Prunus mume belong to the cytoplasmic type.

According to the microstructural characteristics of male sterile flower buds and fertile flower buds at different development stages of Prunus sibirica, anther development can be divided into four stages, namely sporogenous cell stage, tetrad stage, microspore stage and pollen maturity stage. This was similar to the findings reported by Zhang et al. (2018a) and Zhang et al. (2018b). The reason for the male sterility in Prunus sibirica was because the tapetum and middle layer cells failed to degrade sufficiently. The microspore stage was the key stage of anther abortion, whereas the pollen maturity stage was the post-abortion stage.

The soluble sugar content in male sterile flower buds of Prunus sibirica was significantly lower than that in fertile flower buds during the key stage of anther abortion and post-abortion stage, and the difference reached the maximum at the post-abortion stage. This result indicated that the degree of carbohydrate metabolism in male sterile flower buds was lower, which possibly led to a deficiency in the supply of required energy, and therefore, anther abortion (Han et al., 2020). The content of soluble protein in male sterile flower buds of Prunus sibirica was significantly lower than that of fertile flower buds during the key stage of abortion and post-abortion stage, and the difference reached the maximum during the key stage of abortion. The starch content in male sterile flower buds was significantly higher than that in fertile flower buds during the key stage of abortion, and significantly lower than that in fertile flower buds during the post-abortion stage. This result indicated that anther abortion was closely related to the inability of macromolecular nutrients to produce soluble small molecular substances in time, which was consistent with the results of Liu et al. (2014) on male sterility in Sesamum indicum.

In recent years, many studies have used transcriptome sequencing to investigate male sterility, including studies in pepper (Lv et al., 2020), Catalpa bungei (Mao et al., 2017), maize (Xue et al., 2019), and soybean (Li et al., 2019). However, no study has investigated male sterility of Prunus sibirica using transcriptome sequencing. The filtered clean reads account for more than 80% of raw reads (Ge et al., 2014). The GC content can reflect the structure of nucleic acid sequence within a certain range and can be used as an important feature of a sequenced transcriptome (Xu et al., 2020). In our study, the proportion of clean reads obtained from each repeated sample was more than 98%, and the GC content was determined to be between 45.65% and 45.88%. Taken together, these analyses showed that the reliability of the sequencing data was high.

Male sterility is regulated by a series of genes related to fertility, which can be broadly divided into abnormal meiosis genes (Nonomura et al., 2004), abnormal tapetum development genes (Li et al., 2006), abnormal callose metabolism genes (Wan et al., 2011), abnormal pollen wall formation genes (Shi et al., 2011) and abnormal anther cracking genes (Steiner-Lange et al., 2003). The development of the tapetum is closely related to male sterility, and the tapetum is the innermost tissue of the pollen sac wall, which secretes enzymes needed to degrade the callose component of the wall (Cui et al., 2017). Furthermore, the tapetum provides essential nutrients for the development of microspore mother cells (Li et al., 2020), and it is important for normal pollen development. Through the enrichment analysis of differentially expressed genes, the biological functions and metabolic pathways related to male sterility of Prunus sibirica were screened. We found that “protein kinases activity” (Mizuno et al., 2007), “apoptotic process” (Love, Milner & Sadanandom, 2008), “calcium ion binding” (Cao et al., 2012), “cell death” (Zhu et al., 2006), “cytochrome c oxidase activity” (Luo et al., 2013), “aspartic peptidase activity” (Li et al., 2006) and “cysteine- peptidase activity” (Niu et al., 2013) and other terms are closely related to insufficient degradation of tapetum. Biological pathways such as “starch and sucrose metabolism” and “proteasome” also participate in the formation of male sterility of Prunus sibirica. These analyses therefore provided strong initial evidence for elucidating the regulatory pathways that mediate the onset of male sterility in Prunus sibirica.

Conclusions

Based on our results, we suggest that during anther development in Prunus sibirica, starch and other macromolecular substances cannot be converted into soluble small molecules in time due to abnormal metabolic process such as the release of cytochrome c, resulting in the abnormal development of tapetum cells and abnormal pollen development to form male sterility. The stage, characteristics and related metabolic pathways of male sterility in Prunus sibirica were analyzed at the microscopic and transcriptome level in our study, which has important reference value for the study of reproductive development and the mechanism of male sterility in Prunus sibirica. At the same time, this study provides scientific reference for breeding male sterile germplasms with better comprehensive traits, and the selected germplasms can be used as a worthful female parent for hybridization breeding. It is of great significance for giving full play to heterosis and utilizing reasonably the Prunus sibirica germplasm resources in future hybrid breeding.

Supplemental Information

Supplemental Information 1 The volcano plot of differentially expressed genes

Click here for additional data file.

Supplemental Information 2 Differential gene expression profiles involved in programmed death of tapetum

Click here for additional data file.

Supplemental Information 3 Differential gene expression profiles involved in macromolecular nutrients metabolism

Click here for additional data file.

Supplemental Information 4 Sequence information of primers for RT-qPCR

Click here for additional data file.

Supplemental Information 5 Quantitative examination results of total RNA

Click here for additional data file.

Supplemental Information 6 Quality statistics for transcriptome sequencing (RNA-seq) data

Q30: The percentage of bases with Phred value > 20 to the total bases, where Phred =−10Log10e.

Click here for additional data file.

Supplemental Information 7 The summary statistics of the assembled transcripts and unigenes

Click here for additional data file.

Supplemental Information 8 The success rate of functional annotation in transcriptome unigenes

Click here for additional data file.

Supplemental Information 9 Detailed information of differentially expressed genes in GO functional enrichment analysis

Click here for additional data file.

Supplemental Information 10 Physiological indexes between MSFBs and MFFBs

Click here for additional data file.

Supplemental Information 11 Detailed information of differentially expressed genes in KEGG pathway enrichment analysis

Click here for additional data file.

Supplemental Information 12 Detailed information of candidate genes in RT-qPCR

Click here for additional data file.

Supplemental Information 13 The Ct value of sample

Click here for additional data file.

We thank Professor Xiujun Lu and Lecturer Xiaolin Zhang from Shenyang Agricultural University for their helpful comments and suggestions to improve our manuscript.

Additional Information and Declarations

Competing Interests

Author Contributions

Data Availability

The authors declare there are no competing interests.

Jianhua Chen and Hao Xu conceived and designed the experiments, performed the experiments, analyzed the data, prepared figures and/or tables, authored or reviewed drafts of the paper, and approved the final draft.

Jian Zhang conceived and designed the experiments, performed the experiments, prepared figures and/or tables, and approved the final draft.

Shengjun Dong conceived and designed the experiments, authored or reviewed drafts of the paper, and approved the final draft.

Quangang Liu analyzed the data, prepared figures and/or tables, authored or reviewed drafts of the paper, and approved the final draft.

Ruoxi Wang performed the experiments, prepared figures and/or tables, and approved the final draft.

The following information was supplied regarding data availability:

The transcript sequence data is available at NCBI: PRJNA705945.

The assembled data is available at NCBI: GJJH00000000.

Raw data are available in the Supplementary Files.

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
