# Peer review of "Transcriptomic analysis reveals candidate genes for male sterility in Prunus sibirica"

_PeerJ, doi:10.7717/peerj.12349_

## Round 0.1 · original submission · Major Revisions

Dear authors,

Based on the combined comments of the three expert reviewers in this field, my recommendation at this time is for your originally submitted manuscript to undergo major revision prior to resubmission of a significantly improved revised version of your study.

What is also required along with the revised manuscript version is a comprehensive 'Responses to Reviewers' Comments' document which is to detail how the authorship team has addressed each concern raised by each of the three reviewers. Without such a supporting document I will not be able to support a second round of review of the revised manuscript.

·

Basic reporting

Interesting transcriptomic study related to male sterility in Prunus sibirica.

However, some important defects must be revised before publication.

Experimental design

The assyed P. sibirica clone must be described or refernced.

In figure 9 and in the methodology authors must incorporated the correlation coefficient for the statistical evakluation of correlation between RNA-Seq and qPCR analysis.

Tables 3 and 4 are not of great inmteerst. Should be incorporated as supplementary.

Validity of the findings

Conclusions must be improved including main implications of the obtained results from a breeding and production point of view.

Additional comments

Introduction section shopuld be completed with some key refernces about prunus male sterility and Prunus RNA-Seq estudies.

Please clarify objectives of the work in a separated parragrapgh.

Figures 2 5 and 8 should be aded as supplementary material.

Descritpion of Results is very poor. Authors must clarify results showed in the Figures and Tables.

Discussion section should be improved including key refernces about prunus male sterility and Prunus RNA-Seq estudies.

Reviewer 2 ·

Basic reporting

no comment

Experimental design

no comment

Validity of the findings

no comment

Additional comments

Recommendation: Major Revision

In the manuscript “Transcriptomic analysis reveals candidate genes for male sterility in Prunus sibirica”, authors documented morphological characteristics of male sterile flower buds and male fertile flower bus in Prunus sibirica and conducted a RNA-seq analysis on these materials. Male sterility (MS) is an important trait that has been used successfully and widely for plant breeding. This paper addresses an important biology question and authors identified candidate genes responsible for MS in Prunus sibirica. RNA-seq data generated in this study are successfully validated by using qRT-PCR. However, the writing is problematic and improvement is needed. It’s very difficult to see the significance of this research partly due to this reason. What authors present here are just a collection of general descriptive dataset without thoughtful elaboration and meaningful interpretation.
There are major aspects within this manuscript that need to be emended or addressed, authors need to conduct a thorough self-examination according to scientific publication standard.
Specific comments are as following.
1.Introduction
1.1 Authors should give a little information about pollen development and its relationship with the degeneration of the tapetum and middle layer in introduction, so that readers can understand their suggestion that “the reason of male sterility in Prunus sibirica was that tapetum and middle layer cells failed to degrade sufficiently”.
1.2 Relevant references should be cited properly. For instance, L51-55, “the low and unstable yield...affects the yield of Prunus sibirica”, refer to relevant references here. This is true throughout the entire manuscript.
1.3 L61-64, Please update the references. It is not appropriate to use the references published in 2016 or even earlier to represent the work “in recent years”.
1.4 L65-66, “there are few reports on the application of transcriptome sequencing technology to study male sterility”, this is clearly not the case.
1.5 L71, biological functions were not investigated in this work. Authors only annotated DEGs in 7 database. Please rephrase.
2.Materials & Methods
2.1 I am concerned about the genetic backgrounds of clone No. 1 and No. 60, because I consider that there is no scientific meaning in comparing two materials with different genetic backgrounds.
2.2 L91-93, what do the authors mean with “MSFB” and “MFFB”? Are they used to represent male sterile clone No. 1 and male fertile clone No. 60, respectively? Please explain what do they mean.
3. Results
3.1 Fig. 1 are very low-resolution and it is very hard to read. The color of stained anthers in Fig. 1C was not uniform with other panels. It would be much easier to read Fig. 1 if the length of scale bars is uniform in the two columns when showing male sterile flower buds and male fertile flower bud. It is very tedious to check the differences between Fig 1C and G at microspore stage, Fig 1D and H at anther maturity, particularly when the magnification of panels in the two columns is different. This makes it even more difficult to interpret the figure.
The legend of Fig. 1 is unclear. “A, E: sporogenous cell stage; B, F: tetrad stage; C, G: microspore stage; D, H: anther maturity stage.... A, D: sporogenous cell stage; B, F: tapetum stage.” please correct.
Fig. 1D is similar with Fig. 1G. How do you judge that the pollen in Fig. 1D is at anther maturity stage rather than microspore stage as showed in Fig. 1G.
3.2 Instead of showing morphological characterization of male fertile and male fertile plants using paraffin section, it would be clearly more informative to provide evidence using semi-thin sections (1 μm) stained with toluidine blue.
3.3 L170-177, “In male fertile flower bud...(Fig. 1C, D). In male sterile flower buds...(Fig. 1G, H)”. However, in Fig. 1, A-D showed the microstructure of male sterile flower buds and E-H represented male fertile flower bud. Please refer to the correct subgraphs.
3.4 L178-179, “...was one of the causes of male sterility...”, do you mean there are other causes? What are they?
3.5 L189, please explain the critical stage. Isn’t this the “microspore stage”?
3.6 L200, describe the specific databases, not simply mention as “7 databases”.
3.7 L209-210, “Gene Ontology (GO) functional...GO functional classification”, please correct.
3.8 L229-230, “Among them, a total of 297 related genes were enriched.” I don't see any result supports this claim. Please provide more details, either by figure or table.
3.9 L237, please define “4 categories”.
3.10 L246-247, authors mentioned that “these pathways played a key role in regulating the formation of male sterility”. KEGG pathway enrichment analysis alone is not enough to draw such a conclusion. A biological explanation is needed for such a significant result, I don’t see that in the current manuscript.
3.11 Sequencing data should be deposited into public databases, such as NCBI GEO/SRA and include the accession IDs in the manuscript.
4 Discussion
4.1 L275-276, “Male sterile flowers appeared withered and atrophic in full bloom, and pollen sacs can’t crack normally”. I don’t see the description in the Results section of the current manuscript.
4.2 L284-286, the authors stated that “We speculated that part of the proteins regulating anther development were degraded, and the related cellular metabolism process was blocked, thus leading to male sterility”. That statement comes out of the blue without any connection to the data presented.
5 Table
5.1 Table 1
In general, male sterile plants show a reduction in size compared with its male fertile line. However, clone No. 1 shows an increase in tree height, trunk diameter and crown width. This result makes us very curious. Please indicate the genetic backgrounds of clone No. 1 and No. 60.
5.2 Table 2-6 should be included as a supplemental material.
6. The quality of language in the manuscript needs some extensively edited.

Reviewer 3 ·

Basic reporting

The English language should be improved to facilitate understanding of the text. I suggest you have a colleague who is proficient in English and familiar with the subject matter review your manuscript, or contact a professional editing service.
The Introduction contextualizes properly the knowledge gap to be filled. However, the last paragraph needs to be improved with a more realistic perspective regarding the findings of this manuscript.
The literature data are not accessible, since there are too many articles that are not available in the databases or are available only in Chinese, which prevents the access of the international audience. Thus, the references need to be improved by changing the articles unavailable or in Chinese to articles available in English.
The figures and labels need to be revised since some do not properly report the individual names or disagree with the Results.

Experimental design

The research is original, relevant, and extremely meaningful. However, the anatomical analysis conducted was not the most appropriate and prevented that the aims of this study were achieved. Besides that, the anatomical method was based on an article in Chinese and was not sufficiently detailed, which does not permit be replicated. Therefore, a more accurate anatomical analysis needs to be conducted, as suggested in the Material and Methods.

Validity of the findings

The manuscript was revised in order to enrich the quality of the article, and all considerations were made throughout the text in the PDF file.
Unfortunately, the structural analysis conducted precluded an accurate structural characterization of the pollen grain at the different developmental stages and the identification of the tissues that compose the anther, as this manuscript aims. Therefore, the inferences of the tissues and the pollen grain developmental stage related to the male sterility cannot be realized. Besides that, as the transcriptome analysis was based on the results obtained by the structural analysis, both structural and transcriptome data need to be reviewed.

Additional comments

I encourage the authors to resubmit the manuscript at another time. I believe that with new structural analysis, as suggested in the Material and Methods, the pollen grain developmental stage and the tissues involved in the male sterility will be determined and the combination of the ontogenesis data with the transcriptome data will provide important references for the study of the reproductive development and the mechanism of male sterility in this species.

Annotated reviews are not available for download in order to protect the identity of reviewers who chose to remain anonymous.

---

## Round 0.2 · Minor Revisions

Dear authors,

Thank you for all of the excellent work by the authorship team that has obviously gone into the generation of a much improved version of your originally submitted, and peer reviewed manuscript.

I have gone through and edited the revised version of your manuscript. Please find the annotated of the manuscript attached. Please address all of my concerns prior to resubmission of a 'minor' revised version of your revised manuscript.

I do not believe that your manuscript needs to go back out for a second round of peer review assessment, however, I do require you to address each of my concerns in order for your manuscript to be at a level acceptable for publication in PeerJ.

Thanks again for all of your efforts to date. I envisage that the changes I am requesting are all very minor in nature, and therefore, should be achievable in a short timeframe. I look forward to seeing the next version of your study.

Kind regards,
Andrew.

---

## Round 0.3 · Minor Revisions

Dear authors,

Thank you kindly for your excellent work following my initial round of review of your manuscript: your efforts have greatly improved the quality of your study. However, a number of English language issues remain which now need to be addressed.

I have attached an annotated version of your revised manuscript for your reference. This edited version of your revised manuscript identifies all remaining changes that I am requesting that the authorship team now make.

I do believe, however, that once you have addressed the identified issues of this version of your manuscript, your study will be ready for publication acceptance. Please therefore very carefully consider and address each of my identified issues.

Thank you again for the time taken to address my concerns, and please do take the same careful and thorough approach for this round of review.

All the best,

Andrew.

---

## Round 0.4 · Minor Revisions

Dear authors,

Thank you kindly for once again taking the time to carefully revise your manuscript, revisions that have now improved your manuscript to the standard suitable for publication acceptance.

However, an additional review of your manuscript by a Section Editor has identified a number of other oversights which must be addressed before the manuscript can be forwarded further through the system. Specifically, the additional changes being requested are;

1) Assembly statistics: what is the average and median length of the assembled unigenes? What is the N50?

2) The assembled unigenes should also be deposited at NCBI.

3) The annotation information described in the paper should be provided for each unigene (supplemental table).

4) Need more information on the bioinformatics: What programs were used to do the annotation?

5) The text is too small in Figure 2 and Figure 5 - please amend.

Thank you for your continued efforts regarding the improvement of your manuscript. Please make the above-listed changes to your manuscript and resubmit the revised manuscript version along with a response document outlining each requested change that you make.

Kind regards,
Andrew Eamens

---

## Round 0.5 · accepted · Accept

Thank you again to the authorship team for making all changes requested by me in this round of review.

The continued efforts by your team are greatly appreciated and have once again improved the standard of your study.

It is my belief that the current version of your study is now at the standard suitable for publication acceptance.

Thank you again, authors,

Kind regards,
Andrew.